# Pedestrian Attribute Recognition with Graph Convolutional Network in Surveillance Scenarios

**Xiangpeng Song \*, Hongbin Yang and Congcong Zhou**

School of Computer Engineering and Science, Shanghai University, Shanghai 200444, China;
hbyoungshu@staff.shu.edu.cn (H.Y.); zhoucongcong@shu.edu.cn (C.Z.)

\* Correspondence: sxptom@shu.edu.cn; Tel.: +86-18019118350

**Abstract:** Pedestrian attribute recognition is to predict a set of attribute labels of the pedestrian from surveillance scenarios, which is a very challenging task for computer vision due to poor image quality, continual appearance variations, as well as diverse spatial distribution of imbalanced attributes. It is desirable to model the label dependencies between different attributes to improve the recognition performance as each pedestrian normally possesses many attributes. In this paper, we treat pedestrian attribute recognition as multi-label classification and propose a novel model based on the graph convolutional network (GCN). The model is mainly divided into two parts, we first use convolutional neural network (CNN) to extract pedestrian feature, which is a normal operation processing image in deep learning, then we transfer attribute labels to word embedding and construct a correlation matrix between labels to help GCN propagate information between nodes. This paper applies the object classifiers learned by GCN to the image representation extracted by CNN to enable the model to have the ability to be end-to-end trainable. Experiments on pedestrian attribute recognition dataset show that the approach obviously outperforms other existing state-of-the-art methods.

**Keywords:** pedestrian attribute recognition; graph convolutional network; multi-label learning

## 1. Introduction

Video surveillance is a part of our daily life, with the advent of the era of artificial intelligence, intelligent video analytics attaches great importance to the modern city since it can pre-alarm abnormal behaviors or events [1]. In this paper, we mainly focus on the pedestrian in the surveillance system. Human attribute analysis has recently drawn a remarkable amount of attentions by researchers for person detection and re-identification, as well as widely applied in plenty of aspects [2]; besides, pedestrian structured representation can obviously reduce surveillance video storage and improve pedestrian retrieve speed in the surveillance system.

However, there are still plenty of challenges. For one thing, human attributes naturally involve largely imbalanced data distribution. For example, when collecting the attribute "Bald", most of them will be labeled as "No Bald" and its imbalanced ratio to the "Bald" class is usually very large [3]. For another, there are plenty of uncertainties in practical scenarios, the resolution of images may be lower and the human body may be blocked by other things [4]. Furthermore, the collecting of labeled samples is labor-consuming.

Extreme learning machine (ELM) has gained increasing interest from various research fields for certain years. Apart from classification and regression, ELM has recently been extended for clustering, feature selection, representational learning, and many other learning tasks [5]. As an efficient single-hidden-layer feed forward neural network, which has generally good performance and fast learning speed, ELM has been applied in a variety of domains, such as computer vision, energy disaggregation [6], and speech enhancement [7]. In [8], authors proposed a novel pedestrian

detection method using multimodal Histogram of Oriented Gradient for pedestrian feature extraction and extreme learning machine for classification to reduce the detection rate of false positives and accelerate processing speed. The experimental results have proved the efficiency of the ELM based method.

Recently, researchers mostly used convolutional neural network to extract image features due to the rapid development of deep learning. As for multi-label classification, researchers have proposed some approaches based on the probabilistic graph model or recurrent attention model to deal with the problem. It is worth mentioning that attention mechanisms is also a popular method. Inspired by [9], we propose a novel model based on the graph convolutional network to model the correlations between labels. For example, when the label "Long Hair" occurs, the label "Female" is much more likely to show up than the label "Male". Following this idea, we construct an adjacency matrix between labels to deliver this correlation into classifiers, then combine the image representation to produce multi-label loss. Our code is hosted at https://github.com/2014gaokao/pedestrian-attribute-recognition-with-GCN. The contributions of the paper are as follows:

- This paper creatively applies the novel end-to-end trainable multi-label image recognition framework to pedestrian attribute recognition, which to our knowledge, is the first one tackling pedestrian attribute recognition by graph convolutional network.
- Graph convolutional network normally propagates information between nodes based on correlation matrix. So, we design the correlation matrix for GCN in-depth and propose some improvement methods. Finally, we evaluate our method on pedestrian attribute recognition dataset, and our proposed method consistently achieves superior performance over previous competing approaches.

The result of the paper is organized as follows: Section 2 comprehensively introduces related work in pedestrian attribute recognition. Section 3 describes the overall architecture of our model in detail. Section 4 verified the superiority of our method using experiments. Section 5 concludes the paper and states future research directions.

## 2. Related Work

Pedestrian attribute recognition has attracted a lot of attention from researchers, with many efforts dedicated to extending the deep convolutional network for pedestrian attributes recognition. Li et al. [10] proposed and compared two algorithms, where one only uses deep features for binary classification and the other considers the correlations between human attributes, which proves the importance of the modeling attribute relationship. The PGDM (pose guided deep model) [11] explored the structure knowledge of the pedestrian body for person attributes recognition. They used the pre-trained pose estimation model to extract deep features of part of the regions and fused them with the whole image together to improve final attribute recognition, however, the model could not be end-to-end trainable. Besides, many researchers have put a lot of effort in incorporating attention mechanisms and the sequence model in pedestrian attribute recognition. A HydraPlus-Net was proposed by Liu et al. [12] with the novel multi-directional attention modules to train complex features for fine-grained tasks of pedestrian analysis. The experiment showed that the method achieved significant improvements against the prior methods even if the model was hard to understand. Wang et al. [13] proposed to use the sequence-to-sequence model to assist attribute recognition. They first split the whole image into horizontal strip regions and formed region sequences from top to bottom which could help them better mine region dependency for better recognition performance, but the recognition accuracy was not satisfactory. Zhao et al. [14] considered the recurrent neural network's super capability of learning context correlations and the attention model's super capability of highlighting the region of interest on the feature map to propose two models, i.e., recurrent convolutional, which is used to explore the correlations between different attribute groups with the convolutional-LSTM (long short term memory) model and recurrent attention, which takes the

advantage of capturing the interest region, with the results significantly improved. Also, there is never shortage of novel approaches. Dong et al. [15] proposed a curriculum transfer network to handle the issue of less training data. Specifically, they first used the clean source images and their attribute labels online to train the model and then, simultaneously appended harder target images into the model training process to capture harder cross-domain knowledge. Their model was robust for recognizing attributes from unconstrained images taken from-the-wild. Fabbri et al. [16] proposed to use the deep generative model to re-construct super-resolution pedestrian images to deal with the problem of occlusion and low resolution, yet the reconstructed image was not good as expected.

Zhong et al. [17] proposed an image-attribute reciprocal guidance representation method. Due to the relationship between image features and attributes being not fully considered, the author not only investigated image feature and attribute feature together, but also developed a fusion attention mechanism as well as an improvement loss function to address the problem of imbalance attributes. Tan et al. [18] proposed three attention mechanisms including parsing attention, label attention, and spatial attention to highlight regions or pixels against the variations, such as frequent pose variations, blur images, and camera angles. Specifically, parsing attention mainly focuses to extract image features, where label attention pays more attention to attribute features, and spatial attention aims at considering problems from a global perspective, however, they do not fully consider the correlation between attributes. Li et al. [19] proposed to recognize pedestrian attribute by joint visual-semantic reasoning and knowledge distillation while the results remain to be discussed. Han et al. [20] proposed an attention aware pooling method for pedestrian attribute recognition which can also exploit the correlations between attributes. Xiang et al. [21] proposed a meta learning based method for pedestrian attribute recognition to handle the scenario for newly added attributes; semantic similarity and the spatial neighborhood of attributes are not taken into account in this method. In [22], the authors theoretically illustrated that the deeper networks generally take more information into consideration which helps improve classification accuracy. Chen et al. [23] first proposed video-based pedestrian attribute recognition. Their model was divided into two channels: spatial channel extract image features, while the temporal channel took image sequences as input to extract temporal features attached with spatial pooling to integrate the spatial features. Finally, they combed the two channels to achieve attribute classification, but they did not consider the spatial and temporal attention for attribute recognition in videos.

## 3. Approach

In this section, we first introduce some preliminary knowledge of multi-label classification and the graph convolutional network, then we discuss the model in-depth.

### *3.1. Preliminary*

#### 3.1.1. Multi-Label Learning

Traditional supervised learning is prevailing and successful, which is also one of the most studied machine learning paradigms, where each object is just represented by a single feature vector and associated with a single label. But, traditional supervised learning methods have many limitations. In the real world, one object can be represented by many labels and many objects might co-occur in one scenario. The task of multi-label learning is to learn a function which can predict the proper label sets for unseen instances. A naïve way to deal with the multi-label recognition problem is to transform the multi-class into multiple binary-classification problems. However, if a label space contains 20 class labels, then the number of possible label outputs would exceed one million ($2^{20}$). Obviously, we cannot afford the overwhelming exponential-sized output size. So, it is necessary and crucial to capture correlations or dependency among labels which can effectively solve these problems. For example, the probability of an image being annotated with label "Female" would be high if we knew it had

labels "Long hair" and "Skirt". For the multi-label classification algorithms, the following three kinds of learning strategies can be concluded as noted in [24]:

- First-order strategy, which directly transform the multi-class into multiple binary-classification problem. This strategy obviously does not take the correlations into consideration and will cause exponential-sized output size;
- Second-order strategy, which only considers the correlations between each label pair; however, label correlations are more complicated than the second-order strategy in real-world applications;
- High-order strategy, which considers all the label relationships by modeling the correlations among labels. Normally, this strategy can achieve better performance but with higher computational complexity.

### 3.1.2. Graph Convolutional Network

Graph convolutional network is a branch of the graph neural network [25]. It is an emerging field in recent years which originates from the limitations of convolutional neural networks. CNN has developed rapidly these years due to its translation invariance and weight sharing, which started the new era of deep learning [26]. However, the convolutional neural network normally operates on regular Euclidean data-like images and speech, and in other words, convolutional neural networks are not good at operating on non-Euclidean data-like graphs. Therefore, many researchers started to think how to define the convolution on non-Euclidean structures and extract features for machine learning tasks. Advance strategies in graph convolution are often categorized as spectral approaches and spatial approaches. This paper mainly focuses on spectral approaches.

Spectral network was proposed by [27]. The convolution operation is defined in the Fourier domain by computing the eigenvalue decomposition of the graph Laplacian. Given Laplacian:

$$L = I_N - D^{-\frac{1}{2}} A D^{-\frac{1}{2}} = U \Lambda U^T, \tag{1}$$

where $D$ is the degree matrix, $A$ is the adjacency matrix of the graph, and $\Lambda$ is the diagonal matrix of its eigenvalues. Notice that $L$ is a symmetrical and positive semi-definite matrix which means $L$ can be decomposed and each eigenvalue of $L$ is called the spectrum of $L$. Then, researchers define traditional Fourier transform on graph, the graph Fourier transform of a signal $x \in R^N$ is defined as $U^T x$, where $U$ is the matrix of eigenvectors of the normalized graph Laplacian.

The convolution on graph can be defined as the multiplication of a signal $x \in R^N$ with a filter $g_\theta = diag(\theta)$ parameterized by $\theta \in R^N$:

$$g_\theta * x = U g_\theta U^T x. \tag{2}$$

Then researchers focus on $g_\theta$ and deduce that the computational complexity of the above operation is $O(n^3)$, which means this operation may result in intense computational complexity. So, researchers hope to come up with a way equipped with fewer parameters and lower complexity. In [28], the author suggests that $g_\theta$ can be approximated by a truncated expansion in term of Chebyshev polynomials $T_k(x)$ up to $K$-th order:

$$g_\theta * x \approx \sum_{k=0}^{K} \theta_k T_k(\widetilde{L}) x, \tag{3}$$

where $\widetilde{L} = \frac{2}{\lambda_{max}} L - I_N$. $\lambda_{max}$ denotes the largest eigenvalue of $L$. Notice that the approximate operation only requires complexity of $O(K|E|)$ and $K+1$ parameters, where $E$ is the edge numbers of

a graph. Next, [29] limits $K = 1$ and approximates $\lambda_{max} \approx 2$ as well as constrains the parameters with $\theta = \theta'_0 = -\theta'_1$ to simplify the operation, then we can obtain the following expression:

$$g_{\theta'} * x \approx \theta'_0 x + \theta'_1 (L - I_N) x = \theta \left( I_N + D^{-\frac{1}{2}} A D^{-\frac{1}{2}} \right) x, \qquad (4)$$

Finally, using renormalization trick, from a macro perspective, the formula can be expressed as follows:

$$H^{l+1} = \sigma \left( \widetilde{D}^{-\frac{1}{2}} \widetilde{A} \widetilde{D}^{-\frac{1}{2}} H^l W^l \right), \qquad (5)$$

where $H^l$ denotes the nodes feature of $l$-th layer, $W^l$ denotes the parameters to be learned of $l$-th layer, $\sigma(\cdot)$ denotes a non-linear operation.

### 3.2. Architecture of Our Model

The overall framework of our approach is shown in Figure 1.

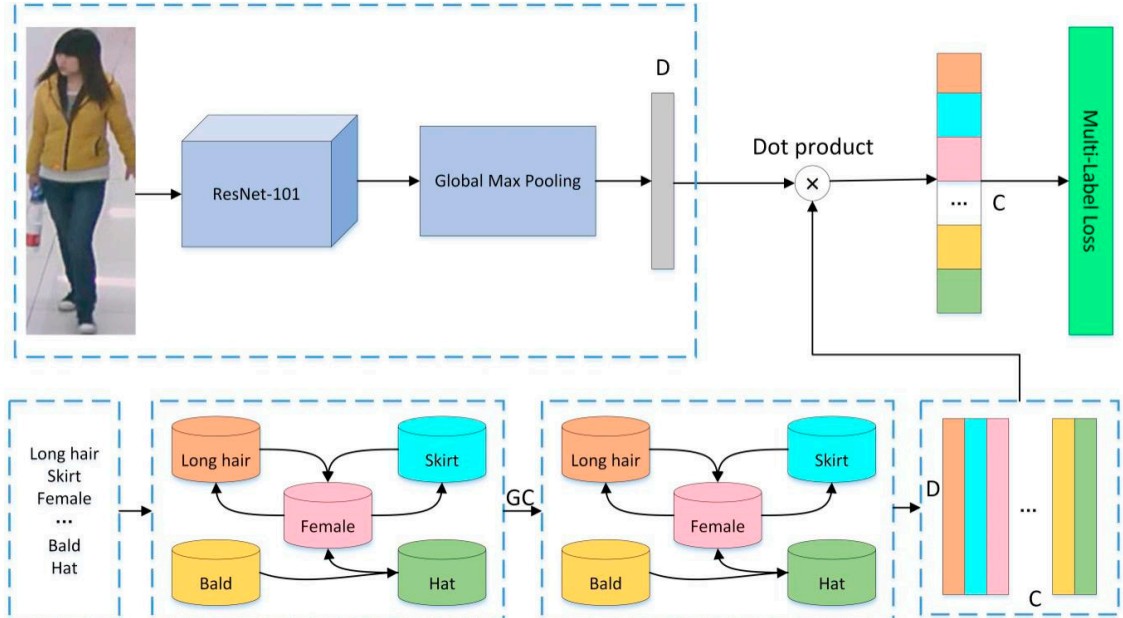

**Figure 1.** Overall framework of our model for pedestrian attribute recognition.

Our model adopts ResNet-101 to extract features of each pedestrian image since ResNet-101 is a common paradigm in image classification, meanwhile, we transform the corresponding attribute labels into word embedding and feed them to our data-driven matrix. The directed line between ellipse pairs represents the dependency of label pairs. Graph convolutional network maps label into D × C-dim classifiers, where D denotes the dimensionality of the parameter to be learned and C denotes the categories of labels. Obviously, our model takes both image and word embedding as input and with multiplication of the corresponding two outputs to produce C-dim scores, and finally, this paper uses traditional multi-label loss to train our network architecture. The detail of image feature extraction and the data-driven matrix is discussed in Sections 3.2.1 and 3.2.2.

### 3.2.1. Image Feature Extraction

Convolutional neural networks have achieved excellent performance in image processing these years; we can use any CNN base models to learn the features of each pedestrian image. In our experiments, this paper uses the deep residual network [30,31] as the base model to push forward our work. Deep residual network originates from the common fault of the deep convolutional neural

network, as many researchers used to stack deeper layers to enrich more features and expected to get better recognition performance. However, is learning better networks as easy as stacking more layers? The answer is absolutely no. As layers go deeper at the beginning stage, accuracy will arise steadily, and as the depth continues increasing, the network may tend to get saturated, then the accuracy degrades rapidly and leads to a higher training error. So, researchers have proposed a solution to the awkward situation as shown in Figure 2. The added layers are identity mapping while the other layers are copied from the learned shallower model.

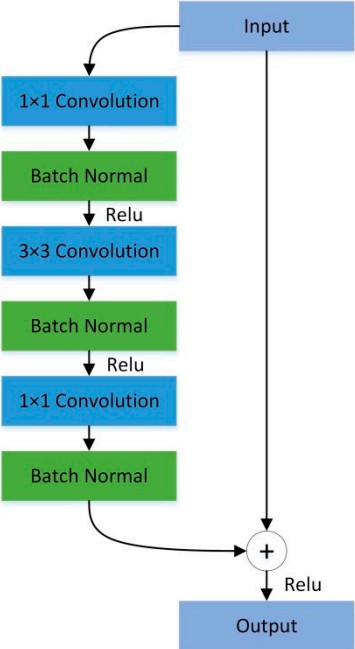

**Figure 2.** Identity mapping.

There are many explanations for why it works. Some researchers suppose that Float 32-bit numbers will no longer represent a gradient change when the network continues backpropagation for a long time once the derivative is less than one at the earlier step, thus shallower networks will not be able to learn things. Instead, deep residual networks take the gradient before derivation into consideration which artificially reduces the probability of gradient dispersion. From my perspective of view, each convolution operation will abandon some information inevitably due to random kernel parameters, inhibition of activation function, and so on. However, residual networks will fetch the information once processed by shortcut to reduce the loss. The authors of the deep residual network have made many model evaluations from 18-layer to 152-layer, and between those are 34-layer, 50-layer, and 101-layer. Our model use ResNet-101 to extract pedestrian image features. We randomly cropped each training image and resized into $224 \times 224$ resolution with horizontal flips for data augmentation. Finally, we can obtain $2048 \times 14 \times 14$ feature maps from the "conv5_x" layer.

### 3.2.2. Correlation Matrix

Graph convolutional network works by propagating information between nodes based on the correlation matrix. Thus, it is crucial to construct a correlation matrix. Normally, we use adjacent matrix to represent the topology structure of a graph. However, such matrix will not be provided in any benchmark pedestrian attribute recognition datasets. In this paper, to construct the correlation matrix, we first define the matrix $N$:

$$\begin{bmatrix} N_1 & N_2 & \cdots & N_{n-1} & N_n \end{bmatrix},$$ (6)

where $N_i$ denotes the occurrence numbers of each label in the training set, then we count the co-occurrence of label pairs in the training set and get the matrix $M$:

$$
\begin{bmatrix}
M_{11} & M_{12} & \cdots & M_{1,n-1} & M_{1n} \\
\vdots & & \ddots & & \vdots \\
M_{n1} & M_{n2} & \cdots & M_{n,n-1} & M_{nn}
\end{bmatrix},
\tag{7}
$$

where $M_{ij}$ denotes the co-occurrence times of $Label_i$ and $Label_j$. Then, we can get the conditional probability matrix by using formula $P_{ij} = M_{ij}/N_i$:

$$
\begin{bmatrix}
P_{11} & P_{12} & \cdots & P_{1,n-1} & P_{1n} \\
\vdots & & \ddots & & \vdots \\
P_{n1} & P_{n2} & \cdots & P_{n,n-1} & P_{nn}
\end{bmatrix}
= \begin{bmatrix}
M_{11} & M_{12} & \cdots & M_{1,n-1} & M_{1n} \\
\vdots & & \ddots & & \vdots \\
M_{n1} & M_{n2} & \cdots & M_{n,n-1} & M_{nn}
\end{bmatrix} / \begin{bmatrix} N_1 & N_2 & \cdots & N_{n-1} & N_n \end{bmatrix},
\tag{8}
$$

where $P_{ij}$ denotes the probability of occurrence of label $L_j$ when label $L_i$ appears. Notice that the formula is not strictly matrix division. The relationship between label pairs is shown as Figure 3, for example, given an image, the probability of occurrence of label "Female" will be very high if label "Skirt" exists, however the probability of occurrence of label "Skirt" might not be high if label "Female" exists, because females wear skirts, but females do not necessarily wear skirts. On the contrary, label "Female" and label "Skirt" both have no relationship with label "Bald", as we seldom see females have no hair.

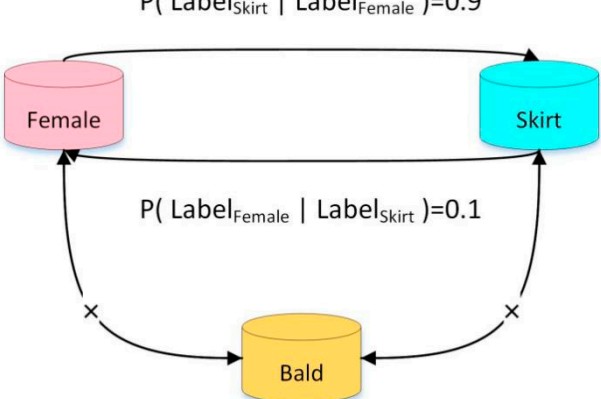

**Figure 3.** Conditional probability between labels.

## 4. Experiments

In this section, we first introduce the evaluation metrics and dataset as well as implementation details. Then, we report the results on benchmark pedestrian attribute recognition dataset.

### 4.1. Evaluation Metrics

Following conventional settings, we record five evaluation criteria including accuracy, precision, recall rate, F1 value, and mA. The formula of mA can be calculated as following:

$$
\mathrm{mA} = \frac{1}{2N} \sum_{i=1}^{L} \left( \frac{TP_i}{P_i} + \frac{TN_i}{N_i} \right),
\tag{9}
$$

where L is the number of attributes. $TP_i$ is the number of correctly predicted positive examples and $TN_i$ is the number of correctly predicted negative examples, $P_i$ is the number of positive examples and $N_i$ is the number of negative examples.

Evaluation metrics on multi-label classification is much more complicated than traditional single-label classification, so there are many evaluation metrics proposed by researchers which can mainly be divided into two groups, i.e., label-based metrics and example-based metrics. Researchers called the above evaluation criterions as label-based evaluation criterions because mA only takes each attribute into consideration and ignores the relationship between attributes. Thus, some researchers suggest using the example-based evaluation criterions like accuracy, precision, recall rate, and F1 value. The formula of those evaluation criteria can be defined as below:

$$Acc_{exam} = \frac{1}{N} \sum_{i=1}^{N} \frac{|Y_i \cap f(x_i)|}{|Y_i \cup f(x_i)|}, \tag{10}$$

$$Prex_{exam} = \frac{1}{2N} \sum_{i=1}^{N} \frac{|Y_i \cap f(x_i)|}{|f(x_i)|}, \tag{11}$$

$$Rec_{exam} = \frac{1}{2N} \sum_{i=1}^{N} \frac{|Y_i \cap f(x_i)|}{|Y_i|}, \tag{12}$$

$$F1 = \frac{2*Prec_{exam}*Rec_{exam}}{Prec_{exam} + Rec_{exam}}, \tag{13}$$

where N is the number of examples, $x_i$ denotes for i-th example, $f(x_i)$ returns the predicted positive labels of $x_i$. $Y_i$ is the ground truth positive labels of the corresponding example.

### 4.2. Dataset

We use RAP-2.0 [32] as our benchmark pedestrian attribute recognition dataset. This dataset contains 84,928 images which were divided into three parts, of which 50,957 for training, 16,986 for validation, and 16,985 for testing. In our experiments, we use both training part and validation part as training dataset. The annotation of each image is very rich as shown in Table 1.

**Table 1.** Annotation attribute names in RAP.

| Class | Attribute |
|---|---|
| Whole body | gender, age, shape, role |
| Head | hair style, hair color, hat, glasses |
| Upper body | clothes style, clothes color |
| Lower body | clothes style, clothes color, shoes style, shoes color |
| Other | bounding box, accessories, actions, occlusion |

### 4.3. Implement Details

We implement ResNet-101 to extract image feature and obtain $2048 \times 14 \times 14$ feature maps from the "conv5_x" layer. Followed by a global max pooling, which size is $14 \times 14$, can we achieve $2048 \times 1 \times 1$-dim image features. Then we select 60 attributes out of 152 in our benchmark dataset and use one-hot word embedding to transform those attribute labels into $60 \times 300$-dim word embedding. As the graph convolutional network normally propagates information based on correlation matrix, we count the total occurrence times of each label in the training dataset to construct matrix $N \in R^{1\times60}$ and we also count the occurrence times of label $L_i$ and label $L_j$ in the training set to construct matrix $M \in R^{60\times60}$, then we can get probability matrix $P = M/N \in R^{60\times60}$ as correlation matrix. Thus, graph convolutional network can work by mapping those label representations to classifiers based on the correlation matrix $P$ and we will get an output vector of $2048 \times 60$ dimensionality.

Obviously, we will get a 60-dim vector by 2048 dot product 2048 × 60. We take the 60-dim vector into traditional multi-label classification loss to ensure our model framework has the ability to be end-to-end trainable.

### 4.4. Experiment Results

As shown in Table 2, our model is a close match against most previous methods before further improvement. To improve final recognition performance, we optimize the correlation matrix, while the details are discussed in Section 4.5. Unsurprisingly, we get a significant improvement according to experiment results. Experiments verified that our improved method goes beyond other competitive counterparts in some of evaluation criteria. Improve operations are discussed in detail in next section.

**Table 2.** Comparison against previous methods.

| Method | RAP | | | | |
|---|---|---|---|---|---|
| Metric | mA | Accuracy | Precision | Recall | F1 |
| ACN | 69.66 | 62.61 | 80.12 | 72.26 | 75.98 |
| DeepMar | 73.79 | 62.02 | 74.92 | 76.21 | 75.56 |
| HydraPlus-Net | 76.12 | 65.39 | 77.33 | 78.79 | 78.05 |
| JRL | 77.81 | - | 78.11 | 78.98 | 78.58 |
| VeSPA | 77.70 | 67.35 | 79.51 | 79.67 | 79.59 |
| Ours | 74.04 | 65.78 | 80.12 | 76.88 | 78.47 |
| Ours-improved | 75.97 | **68.99** | **81.48** | **79.97** | **80.72** |

### 4.5. Ablation Study

In order to further improve final recognition performance, we propose to binarize the correlation matrix $P$, because $P_{ij}$ may be a noisy edge if $P_{ij}$ has a very small value. Specifically, we use the threshold $t$ to filter noisy edges:

$$A_{ij} = \begin{cases} 0, & if\ P_{ij} < t \\ 1, & if\ P_{ij} \geq t \end{cases}.$$ (14)

As shown in Figure 4, it will achieve a better recognition performance than the initial model when the threshold is about 0.6 and obviously it gets top results when the threshold is about 0.9. But it may cause sparse problems, for example, if t = 0.9, then most of edges will be filtered, so we propose to assign proportion $p$, which means the weight between the node itself and its neighbor nodes are:

$$A_{ij} = \frac{p}{\sum_{j=1}^{C} A_{ij} + \varepsilon} A_{ij},$$ (15)

where $\varepsilon$ is a very small number, notice that if the proportion is too small, it will ignore the information from the neighbor nodes; if proportion is too large, it will ignore the feature of the node itself. As shown in Figure 5, our model will get better performance if proportion is close to 0.5, but the proportion strategy does not work for improving overall performance.

Furthermore, we take the GCN layers into consideration to see if the deeper layer can get higher accuracy. We stack more layers to see how the layer numbers affect the model performance. As shown in Figure 6, our model achieves best performance when layer equals to 3. As shown, when the number of graph convolution layers increases over 4, recognition performance drops quickly. The possible reason for the performance drop may be that when using more GCN layers, the propagation between nodes will be accumulated, which can result in over-smoothing.

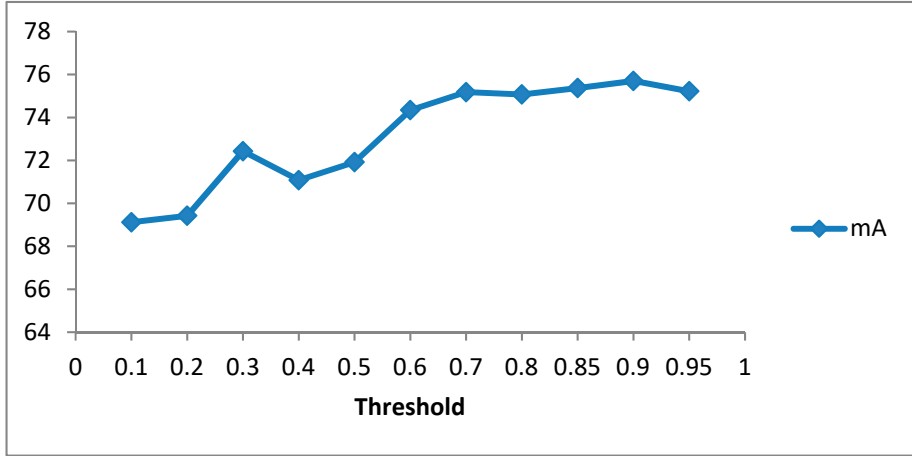

**Figure 4.** Accuracy comparison of different thresholds.

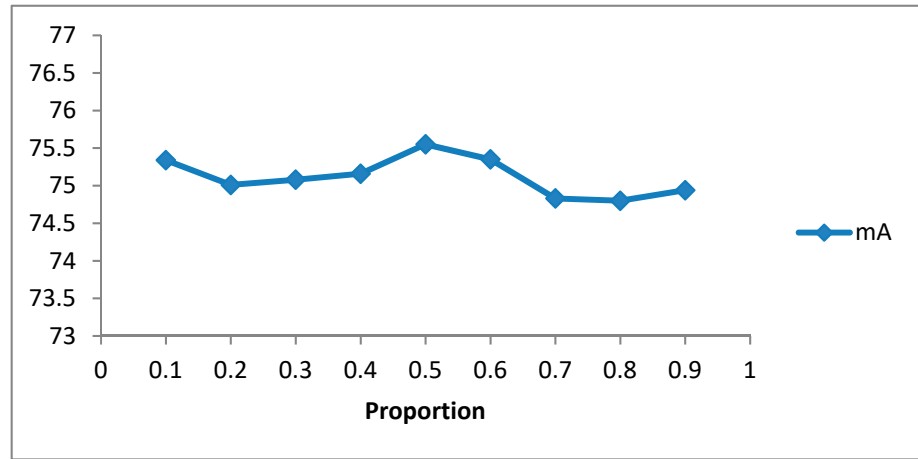

**Figure 5.** Accuracy comparisons of different proportions.

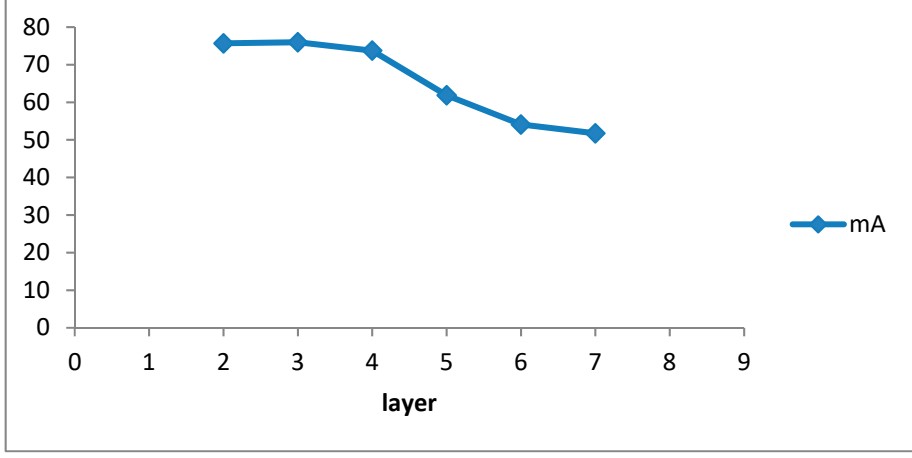

**Figure 6.** Accuracy comparisons of different layers.

*4.6. Image Retrieval*

For a more intuitive display, we wrote a program to retrieve the pedestrian image in test dataset with some query words through our model. As shown in Figure 7, we can clearly observe that our image representation results are pretty accurate.

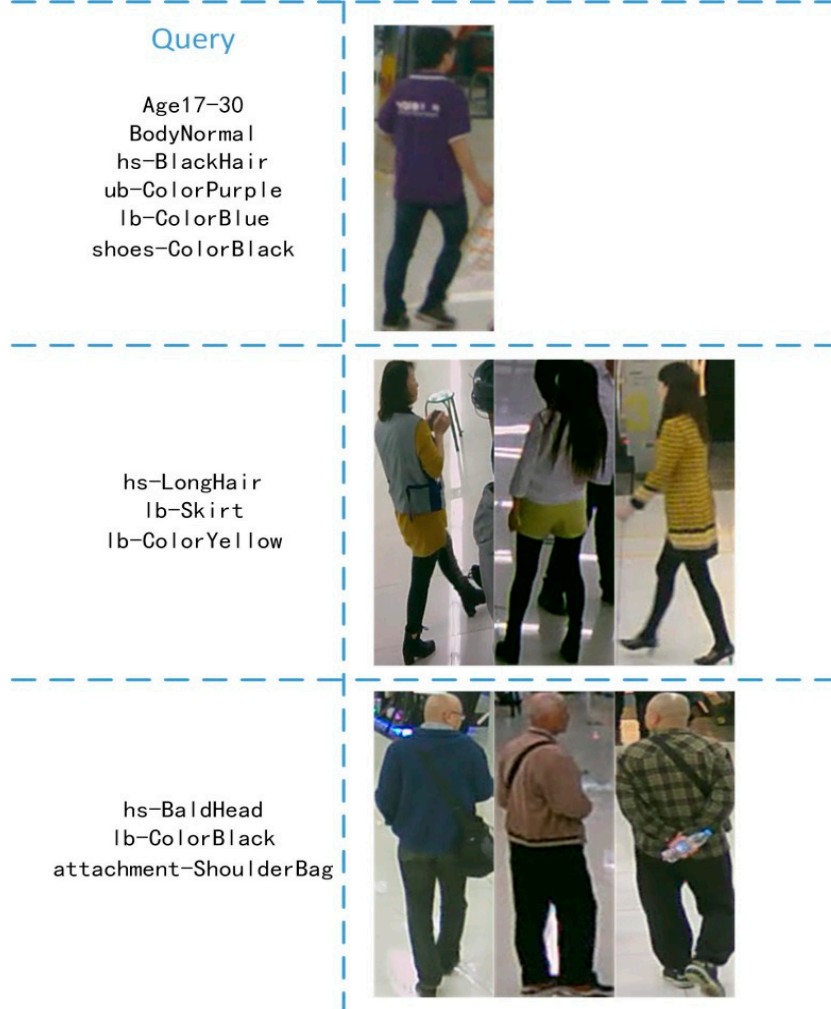

**Figure 7.** Query results.

## 5. Conclusions

Our model overturns conventional approaches, instead we apply a novel framework based on the graph convolutional network for pedestrian attribute recognition and obtain satisfactory results. Given a pedestrian image and corresponding labels, we choose ResNet-101 to extract image features, and transform attribute labels to word embedding, then we construct a correlation matrix according to the occurrence of labels in the training dataset and optimize the matrix for better performance. Graph convolutional network helps us map the information between nodes into classifiers, incorporating the image features extracted by convolutional neural network to ensure our model has the ability to be end-to-end training, which is graceful for network architecture and amazing for final recognition performance. Experiments have validated the superiority of our model.

**Author Contributions:** Conceptualization, C.Z.; software, C.Z.; validation, X.S., H.Y.; formal analysis, X.S.; writing—original draft preparation, X.S.; writing—review and editing, C.Z.

**Funding:** This research received no external funding.

**Acknowledgments:** The authors thank all the anonymous reviewers for their insightful comments and useful suggestions.

**Conflicts of Interest:** The authors declare no conflict of interest.

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
