# Peer review of "Pedestrian Attribute Recognition with Graph Convolutional Network in Surveillance Scenarios"

_futureinternet, doi:10.3390/fi11110245_

Round 1

Reviewer 1 Report

I find the work proposed by the  authors interesting, and sound. Perhaps, the authors may expand their literature review considering that set of networks termed Extreme learning machines (ELMs) that have been proven to be effective in different tasks.

Please discuss and cite the following work in the introduction of your manuscript:

[A] Kai Yang et. al, "An Extreme Learning Machine-based pedestrian detection method," 2013 IEEE Intelligent Vehicles Symposium (IV)

Please refers the following works in the introduction of your manuscript:

[B] Gao Huang, et al. “Trends in extreme learning machines: A review.” Neural Networks 61 (2015): 32-48

[C] Salerno, V.M.; Rabbeni, G. An Extreme Learning Machine Approach to Effective Energy Disaggregation, Electronics, 2018

[D] T Hussain, SM Siniscalchi, CC Lee, SS Wang, Y Tsao, WH Liao. Experimental study on extreme learning machine applications for speech enhancement. IEEE Access 5, 2017

Author Response

    Extreme learning machine is proposed as an efficient single-hidden-layer feedforward neural network which has been proven to be effective in different tasks and has generally good performance and fast learning speed.

    In "An Extreme Learning Machine-based pedestrian detection method", the paper first proposed the prescreening step to reduce the number of slide windows while used location constraints to narrow down the sliding window scanning region and excluded top and bottom part of the frame, followed by edge detection and tree color detection in parallel. After the pre-screening step, they proposed multimodal HOG-based feature extraction in case that a large portion of the hard example have large differences in the leg part. At the training step, they use kernel based ELM to train classifier. Finally, the experimental results outperforms the traditional method in both recognition accuracy and processing speed.

    Also, thank you for your valuable advice, we will discuss it in the introduction of our manuscript and cite the relative work in the references.

Reviewer 2 Report

The authors have introduced a method of multi-label classification for pedestrian attribute recognition with a combination of CNN and GCN, in which the CNN is used for feature extracting in image, the Euclidean Space, while the GCN for graph, the None-Euclidean Space. Instead of traditional NN using for generating correlations between labels, the paper adopt a graph-based NN to deal with the problem. In my view, the GCN method for probabilistic graph between different labels is the main novelty of this paper.

Although the paper is thought-out and well-written, I still have some questions about the description of algorithm and experiments. My questions are as below:

Concerning the explanation for the task of pedestrian attribute recognition, I think a separate section for mathematical description about multi-label attribute recognition and training data is necessary. For example, for an image sampled from training dataset, it contains n targets and features for each target. The first question is about the GCN. The paper use spectral approaches to deal with graph data and the Laplacian matrix is the symmetric normalized Laplacian which is defined with both degree matrix D and adjacency matrix A. In section 3.2.2, in my understanding, P is the adjacency matrix, but I haven’t found any definition for degree matrix D. Is it the diagonal matrix with each element in vector N on its corresponding location of diagonal? So, I think degree matrix D needs more explanation. In section 3.2.2, the conditional probability matrix P is defined by a division with M and N, but there is no division in algorithms of matrix. So the equation (8) should be defined as the matrix multiplication of M and instead of . In section 3.2.2, the conditional probability’s definition is shown in figure 3, in which relationship between female and bald is none, together the same with skirt and bald. But what if a bald man and a woman with skirt occur in the same image. According to the method introduced, the conditional probability for former won’t be zero because of none-zero co-occurrence times in M. I also have some problem with the CNN block. In consideration of the trade-off between speed and precision, we usually adopt simple model trained with ImageNet as backbone at first, such as vgg16 and ResNet18, but the paper use ResNet101 directly. To the best of my knowledge, as for the speed of NNs, the deeper, the slower. So, I think it is important to experiment with simpler backbone as comparison.

Reviewer 3 Report

Pedestrian attribute recognition is a challenging task in surveillance scenarios. The authors proposed a method to deal with this problem.This paper is well organized but not well written. Moreover, there are some problems need to be addressed.

1. Please give the advanages and disadvantages of the related works in Section 2. For example, In the line 65-67, the authors only describe the related work[6], but the description about advantages and disadvantages is not existed.

2. The authors refer the " Euclidean data" in section 3.1.2, please add more explanation about that.

3. The authors introduce the proposed model in section 3.2, more detials about the model and descriptions shoulld be added.

4. In the table 2, the authors show the results of the method (Ours-improved), please add the description about this method? Meanwhile, in the line 282, the authors refers that "we optimize the correlation matrix", please add the more details.

5. Proof reading by a native English speaker is highly recommended. Many statements are not formal.

Round 2

Reviewer 1 Report

The authors have addressed all of my concerns. Thank you.

Reviewer 2 Report

The author has answered all the questions of the first reviewer comments.

Reviewer 3 Report

Authors have addressed the problems that I've concerned.

But many informal statements and grammer errors should be double checked.